# Validation of the Diabetomics CovAB SARS-CoV-2 antibody test in children: comparison with serology

Manisha Gurumurthy,[1] Joseph Schwab,[1] Sri Ram Pentakota,[1] Rahul Ukey,[1] Maria Gennaro,[1] Pauline Thomas,[1] Stephen Friedman[1]

**ABSTRACT**   Monitoring antibody prevalence is a valuable tool to evaluate the burden of severe acute respiratory syndrome coronavirus 2 (SARS-CoV-2) in a community, identify risk factors, and assess the impact of clinical and public health intervention strategies. The antibody prevalence of SARS-CoV-2 in children in the United States in early 2022 was estimated by the Centers for Disease Control and Prevention to be 74.2%, using seroprevalence from a variety of sources. A study by the New Jersey Department of Health in late 2022/early 2023 in unvaccinated children found a lower prevalence, 68% when using a gum swab method to detect antibodies. This study compared the accuracy of the gum swab method to detect antibodies with simultaneously obtained serological samples in additional children. This cross-sectional study recruited well children, not vaccinated for SARS-CoV-2, aged 18 months to 11 years, who were scheduled for routine bloodwork at an inner-city university-based pediatric clinic. With parental consent, an extra 5 cc of blood and a gum swab sample were collected. Results from Diabetomics CovAb SARS-CoV-2 gum swab antibody test and Rutgers New Jersey Medical School enzyme-linked immunosorbent assay serology test for spike protein antibody were compared. The seropositivity of these paired samples was compared using McNemar's test, Cohen's kappa statistic, and other diagnostic accuracy statistics. From June through August 2023, 86 children were recruited. Antibody positivity by gum swab was 70.9% and by serology was 87.2%. The Cohen's kappa statistic was 0.39 indicating minimal agreement and McNemar's test was significant ($P$-value of 0.0010). Compared with serology, gum swab was 78.7% sensitive (95% CI 68.7% to 87.3%) and 81.8% specific (95% CI 48.2% to 97.7%). Positive and negative predictive values were 97.5% and 29.9%, respectively, and accuracy was 79.0%. Sensitivity in non-Hispanic versus Hispanic children was 74.2% versus 82.5%, and in children 6–11 years versus 18 months to 5 years, it was 74.2% versus 81.8%. While the gum swab method of antibody detection is not as sensitive or specific as serology, sample collection can be done in settings where phlebotomy is not feasible. This method could be useful in non-clinical settings such as surveillance, for assessing epidemiological trends and associations.

**IMPORTANCE**   Recently a study determining the prevalence of severe acute respiratory syndrome coronavirus 2 (SARS-CoV-2) antibodies in unvaccinated children in NJ (Katic et al. 2023. Pediatric Academic Societies Meeting; Washington, D.C. https://2023.pas-meeting.org/searchbyposterbucket.asp?bm=Public+Health+%26+Prevention&t=Public+Health+%26+Prevention&pfp=Track) was conducted using a gum swab method for antibody detection. The Diabetomics CovAB test, which qualitatively identifies antibodies to SARS-CoV-2 spike protein, is a point-of-care, low-cost test, that is easy to administer in children. While this test provides sensitive and specific results in adults [US Food and Drug Administration (FDA). 2022. Center for devices and radiological health. EUA authorized serology test performance. Available from: https://www.fda.gov/medical-devices/covid-19-emergency-use-authorizations-medical-devices/eua-authorized-serology-test-performance], data on its accuracy in

Address correspondence to Manisha Gurumurthy, manishabg@gmail.com.

The authors declare no conflict of interest.

See the funding table on p. 6.

children is lacking. As a follow-up to the above-mentioned study, we compared the results of the gum swab test to a serologic antibody test. We found that the gum swab test was inferior to serology but was fairly sensitive and specific with a high positive predictive value. While the test is not ideal for diagnostic purposes in children it can be a valuable tool for public health officials and pediatricians to understand the extent of past health interventions.

**KEYWORDS**    SARS-CoV-2 antibody, children, oral swab testing

As of February 2022, the Centers for Disease Control and Prevention (CDC) estimated the seroprevalence of severe acute respiratory syndrome coronavirus 2 (SARS-CoV-2) in the United States (US) children birth to 11 years increased to 74.2% (1). Studies of antibody prevalence may provide information about the underlying infection rate, especially since coronavirus disease 2019 (COVID-19) in children is often mild or asymptomatic. Monitoring antibody prevalence may guide public health policies to curb the spread of the virus like decisions to enhance vaccination efforts (2, 3).

Most available methods to detect antibodies against SARS-CoV-2 require serum, collected by trained phlebotomists using personal protective equipment. This testing is invasive, expensive, and sometimes poorly accessible (4). Antibody detection using gum exudate or saliva as the biospecimen is recently available and may be particularly useful in children. The Diabetomics CovAb SARS-CoV-2 Ab test is a point-of-care test that uses a gum swab to collect an oral specimen to detect antibodies to the SARS-CoV-2 spike protein. The US Food and Drug Administration authorized it for use in adults only in 2021 under an Emergency Use Authorization, and the manufacturer reported that the sensitivity and specificity in adults for detecting total (IgG, IgA, IgM) antibodies to the viral Spike protein (S) is 97.1% (95% CI 90.0%; 99.2%) and 97.4% (95% CI 91.1%; 99.3%), respectively, provided that testing occurs >15 days from the onset of symptoms or date of a positive Polymerase Chain Reaction test to SARS-CoV-2 (5, 6). However, validation of this test in children has not been reported to date. When used to determine the antibody prevalence of SARS-CoV-2 in children, the prevalence reported was 68%, which was lower than the 74.2% reported by the CDC (7). One explanation for this is that the gum swab test is less sensitive in children than in adults. This study compares the qualitative results obtained using the Diabetomics CovAb SARS-CoV-2 Ab test, with the qualitative results in a concomitantly collected serum sample.

## MATERIALS AND METHODS

### Sample

This was a convenience sample of children aged 18 months to 11 years who attended a primary care clinic at Rutgers New Jersey Medical School in Newark, New Jersey (NJ), and required routine bloodwork. Most of these children were covered by Medicaid or charity care. The children were residents of NJ, unvaccinated against SARS-CoV-2, and had no symptoms of respiratory tract infection on the day the test was administered.

### Sample size

Sample size calculations were performed for McNemar's test to allow for comparison of binary outcomes from paired gum swabs and serology samples. Seroprevalence was assumed to be 90%, power of 80%, significance of 5% for one-sided hypotheses, and difference in positivity proportion between gum swab and serology was assumed to be −15% (mid-point of −10% to −20%). Using the tables published by Agoklu, under a scenario of minimal and maximal chance agreement, between 52 and 93 subjects, with a mean of 73 were needed (8). Yate's continuity corrections were already applied to these calculations.

## Data collection and laboratory analysis

Study enrollment occurred between June and August of 2023. Gum exudate collection and phlebotomy for serology were done at the same clinic visit. After parental consent, gum exudate was tested using the Diabetomics CovAb SARS CoV-2 antibody test. The top and bottom gums were swabbed twice using a small soft paddle. The paddle was then soaked in the provided solution and five drops of this solution were put on a cartridge, left for 15 minutes, and then examined by eye to see if there was a pink or red line indicating the presence of antibody.

The serum samples were analyzed using an enzyme-linked immunosorbent assay protocol in which antibody binding to the Receptor Binding Domain of the S1 subunit of the viral S protein is reported as positive or negative (9).

In addition to the biological specimens, the child's age, sex, race/ethnicity, and history of prior COVID-19 illness were collected.

## Statistical analysis

All analyses were performed using SAS v.9.4 and MedCalc's online "Diagnostic test evaluation calculator" (10, 11). Analyses included cross-tabulations, performing McNemar's test and yielding Cohen's kappa statistic with 95% confidence intervals, computation of diagnostic accuracy statistics, such as sensitivity, specificity, positive and negative predictive values, positive and negative likelihood ratios, and accuracy along with their 95% confidence intervals. These analyses were performed in the overall study cohort and within sub-groups by age (under 5 years versus 5 years to 11 years) and ethnicity (Hispanic versus non-Hispanic). The association was considered significant at the $P < 0.05$.

## RESULTS

Gum swabs and serum samples were collected from 87 children within the study period. However, one child had previously been vaccinated and did not match the inclusion criteria. Data from 86 children were included in the final analysis. The mean age of children was 5.1 years (SD = 2.9); the median was 4.5 years. Fifty (58.1%) participants were male, and 36 were female (41.9%). All but two participants were either non-Hispanic black (37 children, 43.0%) or Hispanic (47 children, 54.7%). Two were Asian. Most (71 children, 82.6%) reported not having a prior history of COVID-19 (Table 1).

Sixty-one (70.9%) children were positive for antibodies to S protein using the gum swab test. Comparatively, 75 (87.2%) serum samples collected from the same children were positive for antibodies to the S protein (Table 1). The sensitivity, specificity, accuracy, and positive/negative predictive value of the gum swab test to detect antibodies were measured against the serology results as the gold standard (Table 2).

The sensitivity of the gum swab test was 78.7% (95% CI: 68.7%–87.3%) and the specificity was 81.8% (95% CI: 48.2%–97.7%). At the assumed seropositivity of 90%, the positive and negative predictive values of the test were 97.5% (95% CI: 91.7%–99.3%) and 29.9% (95% CI: 20.3%–41.7%), respectively. The accuracy was determined to be 79%. The *P*-value of McNemar's test was 0.0010 indicating that the seropositivity proportion between serology and gum swab test results were statistically significantly different. The Cohen's Kappa statistic was 0.39 (95% CI: 0.18–0.60) thereby demonstrating that the degree of agreement in the seropositivity results from serology and gum swab tests was minimal.

Given the possibility that the two large groups in our sample, the Hispanic and non-Hispanic children, had different exposures to SARS-CoV-2, we compared them to see if any differences emerged. The sensitivity of the gum swab test in the non-Hispanic group was 74.3% (95% CI: 57.7%–87.5%) and in the Hispanic group was 82.5% (95% CI: 67.2%–92.7%) (Table 3), but this difference was not statistically significant. Similarly, the sensitivity of the test was 81.8% (95% CI: 67.2%–91.8%) in children 18 months to 5 years as compared to 74.2% (95% CI: 55.4%–88.1%) in children 6 to 11 years (Table 4) and this difference was not statistically significant.

**TABLE 1** Demographic information and test results: *n* = 86

| Variable | Category | Number | Percent |
|---|---|---|---|
| Age in years | ≤5 | 52 | 60.5 |
| | 6–11 | 34 | 39.5 |
| Sex | Male | 50 | 58.1 |
| | Female | 36 | 41.9 |
| Race/ethnicity | Non-Hispanic Black | 37 | 43.0 |
| | Non-Hispanic Asian | 2 | 2.3 |
| | Hispanic | 47 | 54.7 |
| History of COVID-19 | Yes | 15 | 17.4 |
| | No | 71 | 82.6 |
| Gum swab test result | Positive | 61 | 70.9 |
| Serological test result | Positive | 75 | 87.2 |

## DISCUSSION

This study compared the results of antibodies to the SARS-CoV-2 S protein obtained using a gum swab to those obtained using a serum sample in children aged 18 months to 11 years. The sensitivity of the Diabetomics CovAb SARS-CoV-2 Ab test to detect antibodies to the SARS-CoV-2 S protein in our sample was 78.7% (95% CI: 68.7%–87.3%), and its specificity was 81.8% (95% CI: 48.2%–97.7%). The positive predictive value of the test was high (0.97). Our cohort of mainly Hispanic, and non-Hispanic Black children had high antibody prevalence for SARS-CoV-2, regardless of a prior history of COVID-19 illness. Differences were also noted in the sensitivity and specificity of the test between the groups 18 months to 5 years and 6 to 11 years and between the Hispanics and non-Hispanics, but these were not statistically significant. The degree of agreement in the seropositivity results from serology and gum swab methods was minimal with the Cohen's Kappa statistic calculated as 0.39 (95% CI: 0.18–0.60). This finding could partially be explained by Thompson and Walter's work on the strong dependence of kappa on true prevalence (12). When disease prevalence is high, as in the case of COVID-19, the positive predictive value of a test increases, but the negative predictive value decreases, and vice versa. Thus, a negative result from the oral swab test may be less reliable, leading to lower agreement between the two tests. The sensitivity of the oral swab test and hence the agreement between the two tests could also be influenced by the timing of the test with respect to a SARS-CoV-2 infection. There is limited data on the comparison of SARS-CoV-2 antibody levels in oral secretions versus serum, as humoral immunity to the infection wanes, especially in the pediatric population, and is an area of active research. Serological tests for antibody detection are likely more sensitive in this scenario. Collection of the gum swab sample, while straightforward, may be prone

**TABLE 2** Contingency table to assess the sensitivity and specificity of the gum swab test versus serology: *n* = 86

| Gum swab test by the serological test | | | |
|---|---|---|---|
| **Gum swab test result** | **Serological test result** | | |
| | **Positive** | **Negative** | **Total** |
| Positive | 59 | 2 | 61 |
| Negative | 16 | 9 | 25 |
| Total | 75 | 11 | 86 |
| **Statistic** | **Value %** | | **95% CI** |
| Sensitivity | 78.7 | | 67.7 to 87.3 |
| Specificity | 81.8 | | 48.2 to 97.7 |
| Positive predictive value | 97.5 | | 91.7 to 99.3 |
| Negative predictive value | 29.9 | | 20.3 to 41.7 |
| Accuracy | 79.0 | | 68.9 to 87.0 |
| Estimated disease prevalence 90% | | | |

**TABLE 3** Sensitivity and specificity of the gum swab test versus serology among Hispanic versus non-Hispanics: *n* = 86

| Gum swab test result by serology test result | | | | | | | |
|---|---|---|---|---|---|---|---|
| **Hispanic** | | | | **Non-Hispanic** | | | |
| Gum swab test result | Serological test result | | | Gum swab test result | Serological test result | | |
| | **Positive** | **Negative** | **Total** | | **Positive** | **Negative** | **Total** |
| Positive | 33 | 1 | 34 | Positive | 26 | 1 | 27 |
| Negative | 7 | 6 | 13 | Negative | 9 | 3 | 12 |
| Total | 40 | 7 | 47 | Total | 35 | 4 | 39 |
| *n* = 47 | | | | *n* = 39 | | | |
| **Statistic** | **Value %** | **95% CI** | | **Statistic** | **Value %** | **95% CI** | |
| Sensitivity | 82.5 | 67.2 to 92.7 | | Sensitivity | 74.3 | 56.8 to 87.5 | |
| Specificity | 85.7 | 42.1 to 99.6 | | Specificity | 75.0 | 19.4 to 99.4 | |
| Positive predictive value | 98.1 | 89.4 to 99.7 | | Positive predictive value | 96.4 | 82.9 to 99.3 | |
| Negative predictive value | 35.2 | 20.7 to 53.2 | | Negative predictive value | 24.5 | 12.7 to 41.7 | |
| Accuracy | 82.8 | 69.0 to 92.2 | | Accuracy | 74.4 | 57.9 to 87.0 | |
| Estimated disease prevalence 90% | | | | | | | |

to variability in the amount of gum fluid collected. Some children may pull away during swabbing, and some may have more saliva, possibly diluting the sample. Also, one guideline for the use of the Diabetomics CovAb SARS-CoV-2 Ab test is that the subject should not have anything to eat or drink 30 minutes prior to testing. Incorrect adherence to this requirement by staff or the parent or guardian could have affected the accuracy of the results. User errors may also have occurred while reading the test result, most notably a very faint "test line" may have been read as negative, thus undermining test accuracy. Rigorous implementation of the test procedures that include improved staff training and instituting internal controls and monitoring may mitigate some of these errors.

There remains a risk of overgeneralizing the findings, given the real-world utility of a gum swab test to determine SARS-CoV-2 antibody prevalence. The sample size in this study was small and may not be representative of other populations in the United States. Studies with larger sample sizes and diverse populations can strengthen the generalizability of these findings.

The gum swab method is a point-of-care, low-cost test that is easier than serology to administer, especially in children. The test provided fairly good sensitivity and specificity. Since the reported antibody prevalence of SARS-CoV-2 in the general population was reported as approximately 75% as of early 2022 (predating vaccination), the positive

**TABLE 4** Sensitivity and specificity of the gum swab test versus serology between children 18 months to 5 years and 6 to 11 years: *n* = 86

| Gum swab test result by serology test result | | | | | | | |
|---|---|---|---|---|---|---|---|
| **Age 18 months to 5 years** | | | | **Age 6 to 11 years** | | | |
| Gum swab test result | Serological test result | | | Gum swab test result | Serological test result | | |
| | **Positive** | **Negative** | **Total** | | **Positive** | **Negative** | **Total** |
| Positive | 36 | 1 | 37 | Positive | 23 | 1 | 24 |
| Negative | 8 | 7 | 15 | Negative | 8 | 2 | 10 |
| Total | 44 | 8 | 52 | Total | 31 | 3 | 34 |
| *n* = 47 | | | | *n* = 39 | | | |
| **Statistic** | **Value %** | **95% CI** | | **Statistic** | **Value %** | **95% CI** | |
| Sensitivity | 81.8 | 67.3 to 91.8 | | Sensitivity | 74.2 | 55.4 to 88.1 | |
| Specificity | 87.5 | 47.4 to 99.7 | | Specificity | 66.7 | 9.4 to 99.2 | |
| Positive predictive value | 98.3 | 90.4 to 99.7 | | Positive predictive value | 95.3 | 80.0.0 to 99.0 | |
| Negative predictive value | 34.8 | 21.3 to 51.3 | | Negative predictive value | 22.3 | 9.6 to 43.8 | |
| Accuracy | 82.4 | 69.3 to 91.6 | | Accuracy | 73.4 | 55.5 to 87.0 | |
| Estimated disease prevalence 90% | | | | | | | |

predictive value of this test is likely to remain high. Although the test may not be ideal for diagnostic purposes in children or when the antibody prevalence is low, it can be useful for screening, research, or monitoring of antibody prevalence in various settings including the home, clinics, testing centers, remote/rural areas, and community-based screening programs. It can be a valuable tool for public health officials and pediatricians to understand the extent of past infections and for timely public health interventions (4).

## ACKNOWLEDGMENTS

Isaura Otero MPH; Sukhwinder Singh Ph.D.; Bozena Katic Ph.D.

This project was supported by funding from the New Jersey Department of Health.

## AUTHOR AFFILIATION

[1]Rutgers New Jersey Medical School, Newark, New Jersey, USA

## AUTHOR ORCIDs

Manisha Gurumurthy  http://orcid.org/0000-0002-7778-9658
Maria Gennaro  http://orcid.org/0000-0003-4801-3567
Pauline Thomas  http://orcid.org/0000-0001-6853-2837
Stephen Friedman  http://orcid.org/0000-0003-0204-6291

## FUNDING

| Funder | Grant(s) | Author(s) |
| --- | --- | --- |
| State of New Jersey \| State of New Jersey Department of Health (NJDOH) | | Manisha Gurumurthy |

## AUTHOR CONTRIBUTIONS

Manisha Gurumurthy, Conceptualization, Data curation, Funding acquisition, Investigation, Methodology, Project administration, Resources, Supervision, Validation, Visualization, Writing – original draft, Writing – review and editing | Joseph Schwab, Conceptualization, Data curation, Investigation, Methodology, Project administration, Supervision, Validation, Visualization, Writing – review and editing | Sri Ram Pentakota, Conceptualization, Data curation, Formal analysis, Methodology, Software, Writing – review and editing | Rahul Ukey, Conceptualization, Investigation, Methodology, Writing – review and editing | Maria Gennaro, Conceptualization, Investigation, Methodology, Writing – review and editing | Pauline Thomas, Conceptualization, Data curation, Funding acquisition, Investigation, Methodology, Project administration, Resources, Supervision, Validation, Visualization, Writing – review and editing | Stephen Friedman, Conceptualization, Data curation, Formal analysis, Investigation, Methodology, Project administration, Software, Supervision, Validation, Visualization, Writing – review and editing

## ETHICS APPROVAL

The protocol for the study was reviewed and approved by the Rutgers Institutional Review Board. Funding for the study was provided by the NJ Department of Health. Diabetomics, Inc. provided the oral swab testing kits. The authors have no affiliations or financial or non-financial interests with Diabetomics, Inc. and the subject matter and materials discussed in this manuscript are the results of independent research.

## ADDITIONAL FILES

The following material is available online.

Open Peer Review

**PEER REVIEW HISTORY (review-history.pdf).** An accounting of the reviewer comments and feedback.

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
