## [Reviewer comments · Microbiology Spectrum]

Microbiology Spectrum

Validation of the Diabetomics Covab SARS-CoV-2 Antibody Test in Children: Comparison with Serology

Manisha Gurumurthy, Joseph Schwab, Sri Ram Pentakota, Rahul Ukey, Maria Laura Gennaro, Pauline Thomas, and Stephen Friedman

Corresponding Author(s): Manisha Gurumurthy, Rutgers New Jersey Medical School

Review Timeline:

Submission Date:	March 14, 2024
Editorial Decision:	May 7, 2024
Revision Received:	July 31, 2024
Accepted:	August 30, 2024

Editor: Heba Mostafa

Reviewer(s): Disclosure of reviewer identity is with reference to reviewer comments included in decision letter(s). The following individuals involved in review of your submission have agreed to reveal their identity: Ami Davidow (Reviewer #1)

Transaction Report:

DOI: <https://doi.org/10.1128/spectrum.00646-24>

Re: Spectrum00646-24 (Validation of the Diabetomics Covab SARS-CoV-2 Antibody Test in Children: Comparison with Serology)

Dear Dr. Manisha B Gurumurthy:

Thank you for the privilege of reviewing your work. Below you will find my comments, instructions from the Spectrum editorial office, and the reviewer comments.

Revision Guidelines

Sincerely,
Heba Mostafa
Editor
Microbiology Spectrum

Reviewer #1 (Comments for the Author):

1. The agreement between the serological and oral swab tests may be low because the prevalence of SARS-CoV-2 is so high in this population (assuming the serological test represents the true prevalence). See Thompson and Walter, 1988, Journal of Clinical Epidemiology.
2. The conclusion that the oral test may be used to assess epidemiological trends is not supported by the analysis. The time

period over which the children were enrolled was too short to assess any trends.

3. What is the purpose of conducting separate analyses of Hispanic versus non-Hispanic groups?

4. Because of the overlap of the 95% confidence intervals for the group-specific sensitivities, one cannot conclude that the sensitivities are "lower." (Similarly, the sensitivities for the two age groups are also not significantly different.)

5. The study indicates that using the oral swab test in children is not to be relied upon. Conclusion: the oral swab test should be used only in adults, assuming that that the EUA is supported by studies in adults.

Reviewer #2 (Comments for the Author):

Table 1: For those children with a history of COVID-19, were both the gum swab and serum testing positive? Were the dates of disease recorded?

Could you speculate why the sensitivity of the test was slightly higher in the younger cohort or do you think this is associated with the largest sample size of that group?

Is there a concern with the lower sensitivity of the gum swabs/diabetomics test being compounded with amount of time from viral exposure? Longitudinal studies would be an interesting next step.

Responses to Reviewer #2:

1. Table 1: For those children with a history of COVID-19, were both the gum swab and serum testing positive? Were the dates of disease recorded?

Response: For those with a history of COVID-19, there were scenarios when both the gum swab and serum tested positive for SARS-CoV-2 antibodies and when the serum tested positive, but the gum swab tested negative for antibodies. We did not see the gum swab test positive and the serum test negative in the setting of a history of COVID-19. The dates of the disease were not recorded consistently, mainly because of the issue with recall.

2. Could you speculate why the sensitivity of the test was slightly higher in the younger cohort or do you think this is associated with the largest sample size of that group?

Response: It is largely a speculation, but the older kids generally had more freedom accessing food and drinks and not requiring assistance from the parent and caregiver to consume it. Hence the rule that “the subject should not have anything to eat or drink for 30 minutes prior to the oral swab test” may not have been correctly adhered to, thus affecting the accuracy of the test results.

3. Is there a concern with the lower sensitivity of the gum swabs/diabetomics test being compounded with amount of time from viral exposure? Longitudinal studies would be an interesting next step.

Response: The sensitivity of the oral swab test and hence the agreement between the two tests could have been influenced by the timing of the test with respect to a SARS-CoV-2 exposure and infection. There is limited data on the presence of SARS-CoV-2 antibodies in oral secretions as humoral immunity to the infection wanes and is an area of active research. Serological tests for antibody detection are likely more sensitive in this scenario. Longitudinal studies graphing the serological and oral secretion antibody response to SARS-CoV-2 exposure and disease will be a valuable step in assessing the usefulness of the oral swab test in monitoring population immunity and assessment of vaccine responses.

Responses to Reviewer #1

1. The agreement between the serological and oral swab tests may be low because the prevalence of SARS-CoV-2 is so high in this population (assuming the serological test represents the true prevalence). See Thompson and Walter, 1988, Journal of Clinical Epidemiology.

Response: Thank you for raising this point that the agreement between the serological test for SARS-CoV-2 antibody detection (which we have assumed as the gold standard for our study) and the oral swab test which can be influenced by the disease burden in the population. According to Thompson and Walter's 1988 paper in the Journal of Clinical Epidemiology, when disease prevalence is high, even tests that have good sensitivity and specificity might show less agreement. When disease prevalence is high, as in the case of COVID-19, the positive predictive value of a test increases, but the negative predictive value decreases, and vice versa. Thus, a negative result from the oral swab test may be less reliable, leading to lower agreement between the two tests.

The sensitivity of the oral swab test and hence the agreement between the two tests could also be influenced by the timing of the test with respect to a SARS-CoV-2 infection. There is limited data on the comparison of SARS-CoV-2 antibodies in oral secretions vs serum as humoral immunity to the infection wanes, especially in the pediatric population, and is an area of active research. Serological tests for antibody detection are likely more sensitive in this scenario.

Please note changes in the manuscript with track changes in Text, under Discussion, lines 224-237 have been made pertaining to this response.

2. The conclusion that the oral test may be used to assess epidemiological trends is not supported by the analysis. The time period over which the children were enrolled was too short to assess any trends.

Response: The study conducted was purely to compare the sensitivity and specificity of the oral swab test to the serological test to detect SARS-CoV-2 antibodies. A cross-sectional study design with enrollment close to the calculated sample size helped us determine that the oral swab test was inferior as compared to the serological test with respect to sensitivity and specificity. The aim of our study was not to assess epidemiological trends. Given that saliva-based antibody tests can be used easily and more frequently compared to the serological test, it could be a useful tool in monitoring population immunity as in large scale surveillance studies to assess epidemiological trends.

Please note changes in the manuscript with track changes in Key Points, under Meaning, lines 32-33 and in Abstract, under Discussion, line 81 have been made pertaining to this response.

3. What is the purpose of conducting separate analyses of Hispanic versus non-Hispanic groups?

Response: The majority of the parents whose children were enrolled in our study identified as Hispanic, and most of the others identified as non-Hispanic black; only two identified as Asian. It is possible that the Hispanic and non-Hispanic children had different exposures to SARS-CoV-2. We combined the latter groups for comparison purposes and to see if any differences emerged.

Please note changes in the manuscript with track changes in Text, under Results, lines 206-208 have been made pertaining to this response

4. Because of the overlap of the 95% confidence intervals for the group-specific sensitivities, one cannot conclude that the sensitivities are "lower." (Similarly, the sensitivities for the two age groups are also not significantly different.)

Response: We agree that differences were noted in our study with group-specific sensitivities, and in the sensitivities between the two age groups, but were not significantly different.

Please note changes in the manuscript with track changes in Abstract, under Results, lines 75-77; Text, under Results, lines 206-213; and in Text, under Discussion, line 224 have been made pertaining to this response.

5. The study indicates that using the oral swab test in children is not to be relied upon.

Conclusion: the oral swab test should be used only in adults, assuming that that the EUA is supported by studies in adults.

Response: The sensitivity of the Diabetomics CovAb SARS-CoV-2 Ab test to detect antibodies in our sample was 78.7% (95% CI 68.7%- 87.3%) and the specificity was 81.8% (95% CI 48.2%-97.7%) when compared to the serological method of antibody detection. Some of this difference could be attributed to user errors and there is scope in improving sample collection methods and sample processing that may have a positive impact on the sensitivity of the oral swab test.

Acceptable levels of sensitivity and specificity depend on the clinical and public health context of the disease in question. Higher test sensitivity must be prioritized for a highly contagious and severe disease, and higher test specificity must be prioritized when false positive findings lead to significant consequences. Given that SARS-CoV-2 transmission is ongoing in our communities, with children having milder COVID-19 symptoms, the authors feel that the lower level of sensitivity and specificity of the test is acceptable for disease surveillance. We agree that the oral swab would not be appropriate for clinical care.

We have addressed this in Text, under Discussion, lines 253-263.

Re: Spectrum00646-24R1 (Validation of the Diabetomics Covab SARS-CoV-2 Antibody Test in Children: Comparison with Serology)

Dear Dr. Manisha B Gurumurthy:

Your manuscript has been accepted, and I am forwarding it to the ASM production staff for publication. Your paper will first be checked to make sure all elements meet the technical requirements. ASM staff will contact you if anything needs to be revised before copyediting and production can begin. Otherwise, you will be notified when your proofs are ready to be viewed.

Sincerely,
S. Wesley Long
Editor
Microbiology Spectrum